# Corticosterone Induces HMGB1 Release in Primary Cultured Rat Cortical Astrocytes: Involvement of Pannexin-1 and P2X7 Receptor-Dependent Mechanisms

**DOI:** 10.3390/cells9051068

**Published:** 2020-04-25

**Authors:** Kazue Hisaoka-Nakashima, Honami Azuma, Fumina Ishikawa, Yoki Nakamura, Dengli Wang, Keyue Liu, Hidenori Wake, Masahiro Nishibori, Yoshihiro Nakata, Norimitsu Morioka

**Affiliations:** 1Department of Pharmacology, Graduate School of Biomedical & Health Sciences, Hiroshima University, Kasumi 1-2-3, Minami-ku, Hiroshima 734-8553, Japan; hisaokak@hiroshima-u.ac.jp (K.H.-N.); azm_0913@icloud.com (H.A.); b154745@hiroshima-u.ac.jp (F.I.); nakayoki@hiroshima-u.ac.jp (Y.N.); ynakata@hiroshima-u.ac.jp (Y.N.); 2Department of Pharmacology, Graduate School of Medicine, Dentistry and Pharmaceutical Sciences, Okayama University, Shikata, Okayama 700-8558, Japan; dengliwang@okayama-u.ac.jp (D.W.); liukeyue@md.okayama-u.ac.jp (K.L.); wake-h@cc.okayama-u.ac.jp (H.W.); mbori@md.okayama-u.ac.jp (M.N.)

**Keywords:** stress, glucocorticoid, HMGB1, astrocyte, neuroinflammation, major depressive disorder

## Abstract

A major risk factor for major depressive disorder (MDD) is stress. Stress leads to the release of high-mobility group box-1 (HMGB1), which in turn leads to neuroinflammation, a potential pathophysiological basis of MDD. The mechanism underlying stress-induced HMGB1 release is not known, but stress-associated glucocorticoids could be involved. To test this, rat primary cultured cortical astrocytes, the most abundant cell type in the central nervous system (CNS), were treated with corticosterone and HMGB1 release was assessed by Western blotting and ELISA. Significant HMGB1 was released with treatment with either corticosterone or dexamethasone, a synthetic glucocorticoid. HMGB1 translocated from the nucleus to the cytoplasm following corticosterone treatment. HMGB1 release was significantly attenuated with glucocorticoid receptor blocking. In addition, inhibition of pannexin-1, and P2X7 receptors led to a significant decrease in corticosterone-induced HMGB1 release. Taken together, corticosterone stimulates astrocytic glucocorticoid receptors and triggers cytoplasmic translocation and extracellular release of nuclear HMGB1 through a mechanism involving pannexin-1 and P2X7 receptors. Thus, under conditions of stress, glucocorticoids induce astrocytic HMGB1 release, leading to a neuroinflammatory state that could mediate neurological disorders such as MDD.

## 1. Introduction

Major depressive disorder (MDD) is one of the most widespread, debilitating mental illnesses, affecting more than 300 million people worldwide [1]. Stress is a major risk factor for psychiatric disorders including MDD [2]. Neuroinflammatory processes, of both the peripheral and central innate immune systems, are evoked by stress [3,4]. Recently, efforts have been made to identify key mediators of stress-induced inflammation in the context of MDD.

Physical, cellular and psychological stress initiates the release of endogenous factors known as danger- or damage-associated molecular patterns (DAMPs) from immune and non-immune cells to promote “sterile inflammation”, the activation of inflammation processes in the absence of exogenous pathogens [5]. High-mobility group box-1 (HMGB1) is known as a ubiquitous, non-histone chromosomal protein which regulates gene transcription and replication and is abundant in the nuclei of most mammalian cells under physiological conditions [6]. When cells and organisms are subjected to physical and psychological stressors, HMGB1 is secreted and functions as a proinflammatory molecule through binding to toll-like receptor 4 (TLR4), TLR5, receptor for advanced glycation end-products (RAGE) and C-X-C chemokine receptor type 4 (CXCR4) [7,8,9]. HMGB1 has been demonstrated as having a role in a number of diseases with an inflammatory component [10]. In addition, HMGB1 is a major regulator of innate immunity and, in response to stress, can induce sterile inflammation as a DAMP [11].

Recent findings suggest that HMGB1 mediates the neural response to inflammation and stress-induced depressive-like behavior [12]. Inescapable foot shock stress increased hippocampal levels of HMGB1, and knockdown of HMGB1 blocked stress-induced increases in cytokine and chemokine, which are believed to mediate depression-like behavior [13]. High HMGB1 concentrations in the hippocampus and serum have been observed in mice with depressive behaviors induced by chronic unpredictable mild stress (CUMS) [14]. Systemic or intracerebroventricular injections of HMGB1 increased immobility time in the tail suspension test [15,16] and forced swimming test [17], and decreased sucrose preference [18], indicators of depressive-like behaviors in mice. These findings indicate that HMGB1-mediated neuroinflammation is involved in stress-induced depressive-like symptoms. However, the mechanism of stress-induced increase of HMGB1 and the cell which actually showed increased HMGB1 is largely unknown.

Astrocytes are the most abundant cells in the central nervous system (CNS). These cells are assumed to play an important role in a variety of functions ranging from synaptogenesis, neurogenesis, gliogenesis, regulation of extracellular ionic concentration, modulation of synaptic transmission, and neurovascular interaction [19,20]. Astrocytes also participate in the immune and inflammatory response [21,22]. Astrocyte dysfunction has been implicated in the etiopathology of MDD [23]. Astrocytes release HMGB1 in response to a number of stimuli, including interleukin (IL)-1β, dibutylyl cAMP, and forskolin [24,25], through a highly defined process. Following cerebral ischemia in rats, sustained levels of HMGB1 were found in brain astrocytes [26]. Thus, astrocytes may be the one of source of extracellular HMGB1 under conditions of stress.

Systemic glucocorticoid levels are increased with sustained stress [27]. A number of studies suggest that glucocorticoids play a pivotal role in stress-induced potentiation of the neuroinflammatory response [11,28,29]. Thus, stress-induced increase in HMGB1 appears to be mediated in part by glucocorticoids. However, whether glucocorticoids stimulate the release of HMGB1 has yet to be demonstrated. The present study determined whether primary cultured rat cortical astrocytes treated with corticosterone, a major stress hormone in rodents, induced the release of HMGB1. The current findings demonstrate that HMGB1 from corticosterone-stimulated astrocytes is released via a specific signaling pathway involving pannexin-1 hemichannels and P2X7 purinergic receptors.

## 2. Materials and Methods

All experiments utilizing animals were conducted in accordance with the “Guidelines for the Care and Use of Laboratory Animals” established by the Japanese Pharmacological Society and Hiroshima University and procedures were reviewed and approved by the Committee of Research Facilities for Laboratory Animal Science of Hiroshima University (A16-127).

### 2.1. Materials

The following reagents, and their sources, were used: corticosterone (Tokyo Chemical Industry CO., LTD., Tokyo, Japan); dexamethasone, A804598, and A438079 (Fujifilm WAKO Pure Chemical Corporation, Osaka, Japan); mifepristone (Cayman, Ann Arbor, MI, USA); carbenoxolone (Merck KGaA, Darmstadt, Germany) and ^10^Panx (Toronto Research Chemicals, North York, ON, Canada). Concentrations of inhibitors and antagonists, which were selected for their respective targets, were based on previous reports [30,31,32,33,34].

### 2.2. Cell Culture

Primary cultured astrocytes, from 1-day-old neonatal Wistar rats, were prepared as previously described [35]. Primary astrocytes were isolated from mixed culture after 3 weeks. Isolated primary astrocytes were plated onto 35 mm dishes (1 × 10^6^ cells/dish) and >95% of the cells were positive for glial fibrillary acidic protein (GFAP), an astrocytic-specific marker. After two days, the medium was replaced with serum-free Dulbecco’s Modified Eagle’s Medium (DMEM). The cells were used for experiments on the following day.

We chose a glucocorticoid concentration relevant to the mechanism of stress with the objective of using concentrations commonly used in other in vitro gene expression studies and to mimic the range of corticosterone concentrations observed in the rat brain as measured by microdialysis in rodents undergoing stress [36,37]. Concentrations of 0.01 to 10 µM of corticosterone were used.

### 2.3. Western Blotting

The conditioned media sample was spun at 1000 g for 5 min to remove dead cells, and then an equal volume of conditioned media from an equal number of cells from each treatment group was precipitated with 15% Trichloroacetic Acid (TCA) on ice overnight. The precipitated proteins were pelleted by centrifugation at 13,000 *g* for 5 min. The pellet was washed twice with 100% ethanol and then solubilized in radio immunoprecipitation assay buffer (RIPA buffer) with inhibitors (100 mM Tris-HCl, pH 7.4, 150 mM NaCl, 1 mM EDTA, 1% Triton X-100, 1% sodium deoxycholate, 0.1% sodium dodecyl sulfate (SDS), 20 μg/ml aprotinin, 20 μg/ml leupeptin, 1 mM phenylmethylsulfonyl fluoride, and phosphatase inhibitor cocktail (Nacalai Tesque, Kyoto, Japan)). Total cell samples were also solubilized in RIPA buffer with inhibitors. The lysates of precipitated protein from media and total cells were centrifuged at 13,000 *g* for 10 min at 4 °C and the supernatant was collected. The conditioned media and cells samples were added to 3 × SDS Laemli’s buffer and boiled for 5 min. Equal amounts of protein were separated by 12% SDS-polyacrylamide gel electrophoresis and blotted onto PVDF membranes. Blocking buffer was used to reduce non-specific binding, and the membranes were subsequently incubated with a monoclonal antibody against HMGB1 (1:500, MAB1690, R&D Systems, MN, USA), or monoclonal antibody against β-actin (1:10,000, A5441, Merck KGaA, Darmstadt, Germany) overnight at 4 °C. After washing, the membranes were incubated with a horseradish peroxidase-conjugated secondary antibody (Santa Cruz Biotechnology, Santa Cruz, CA, USA) for 1 h at room temperature. Membranes were then rinsed and incubated with Crarity^TM^ Western ECL substrate (Bio-Rad Laboratories, Inc., Hercules, CA, USA), using Image Quant LAS 4000 (GE Healthcare, Waukesha, WI, USA). The membranes were converted to digital images and protein expression was quantified by densitometric analysis of each immunoreactive band using Science Lab Image Gauge (Fuji Film, Tokyo, Japan). In the current study, HMGB1 appeared as a single band at 30kD.

### 2.4. HMGB1 Enzyme-Linked Immunosorbent Assay (ELISA)

The concentrations of HMGB1 protein in cell-conditioned media were determined using a HMGB1 enzyme-linked immunosorbent assay kit (Shino-Test Co, Sagamihara, Japan) according to the manufacturer’s protocol [38].

### 2.5. Real-Time PCR Analysis

Total RNA from primary astrocytes was prepared by a previously described method [39] and used to synthesize cDNA with MuLV reverse transcriptase (Applied Biosystems, Foster City, CA, USA) and a random hexamer primer. cDNAs synthesized using 1 μg of total RNA in each sample were subjected to real-time PCR assays with specific primers and EXPRESS SYBR^®^ GreenER^TM^ qPCR SuperMixes (Invitrogen, Carlsbad, CA, USA). The expression of HMGB1 mRNA, and glyceraldehydes-3-phosphate dehydrogenase (GAPDH) mRNA were detected by using a pair of primers (HMGB1 forward: AAGCACCCGGATGCTTCT, HMGB1 reverse: GCATTGGGGTCCTTGAAC, GAPDH forward: AGCCCAGAACATCATCCCTG, GAPDH reverse: CACCACCTTCTTGATGTCATC). Real-time PCR assays were conducted using a DNA engine Opticon 2 real-time PCR detection system (Bio-Rad Laboratories, Inc., Hercules, CA). The three-step amplification protocol consisted of 3 min at 95 °C, followed by 40 cycles at 95 °C for 15 s, 60 °C for 30 s, and 72 °C for 30 s. RNA quantities of target genes were calculated using the Ct method. The Ct values of HMGB1 amplification were normalized to that of GAPDH amplification.

### 2.6. Immunocytochemistry

Immunocytochemistry of primary cultured rat cortical astrocytes was performed 24 h after corticosterone treatment. After washing, cells were fixed in 4% paraformaldehyde in PBS for 30 min. Cells were then rinsed with PBS, incubated in a blocking solution of 10% goat serum, 3% BSA, 0.2% Triton X and 0.2% Tween-20 in PBS for 30 min at 24–26 °C, and then incubated with a polyclonal antibody against HMGB1 (catalog # ab18256, 1:1000, Abcam, Cambridge, UK) for three days at 4 °C. After three washes with 0.1% BSA-PBS, cells were incubated at room temperature for 1 h with Alexa Fluor^®^ 546 (catalog # A-11003, 1:1500, Thermo Fisher Scientific, Waltam, MA, USA) and counterstained with DAPI (catalog # D9542, 1:2500; Merck KGaA, Darmstadt, Germany) diluted in 3% BSA-PBS. Cells were washed three times with 0.1% BSA-PBS followed by a final rinse in water. Cells were dried and coverslipped with Fluoro-KEEPER Antifade Reagent (catalog # 12593-64; NACALAI TESQUE, INC., Kyoto, Japan). Cells were then viewed with a BZ-9000 Biorevo all-in-one fluorescence microscope (Keyence, Elmwood Park, NJ, USA). Gain and exposure levels were set for control cultures and kept constant for all other cultures within an experiment. HMGB1 translocation from the nuclear compartment to the cytosol and release from the cell were defined as described previously [40]. Cells with HMGB1-positive staining within the nuclear compartment were considered HMGB1-positive (“nuclear HMGB1-positive staining”). A reduction or absence of HMGB1 staining within the nuclear compartment along with positive HMGB1 staining within the cytosolic compartment was considered to be HMGB1 translocation from the nucleus to the cytosolic compartment (“cytoplasmic HMGB1-positive staining”). The HMGB1 positive cells in nuclear and cytosol compartments were calculated as a percentage of the total number of cells, which was manually determined by counting DAPI-positive nuclei using Image J software (NIH, Bethesda, MD, USA). The average of number of counting DAPI-positive nuclei is about 70 in an individual culture within an experiment.

### 2.7. Statistical Analysis

Data are expressed as the mean ± S.E.M. of at least three independently performed experiments. Statistical analyses of differences between more than two treatment groups were performed with one-way analysis of variance (ANOVA) with pairwise comparison carried out by either Tukey’s honest significant difference (HSD) or Dunnett’s test. Differences between two treatment groups were statistically analyzed with Student’s *t*-test. *p* values less than 0.05 were taken as statistically significant.

## 3. Results

### 3.1. Effects of Corticosterone on HMGB1 Release in Rat Primary Cultured Cortical Astrocytes

Concentrations of 1 µM and 10 µM of corticosterone significantly increased HMGB1 release after 24 h treatment (Figure 1A). A lower concentration of corticosterone tended to increase extracellular HMGB1. Concentrations of corticosterone up to 10 µM were not cytotoxic, indicating that HMGB1 release observed with 1 µM and 10 µM of corticosterone was not a result of nonspecific cytotoxicity or necrosis (Appendix A). Next, time course experiments were conducted. With 1 µM corticosterone treatment, extracellular levels of HMGB1 were significantly increased at 24 h after treatment and were increased for at least 72 h (Figure 1B).

HMGB1 release from rat primary cultured cortical astrocytes following treatment with 1 µM corticosterone was confirmed by ELISA (Figure 1C). Twenty-four hours after incubation with 1 µM corticosterone, significant HMGB1 was measured in the media, compared to incubation with vehicle treatment.

### 3.2. Effects of Dexamethasone on HMGB1 Release in Rat Primary Cultured Cortical Astrocytes

Dexamethasone is a selective glucocorticoid receptor agonist and is often used to confirm receptor-mediated effects of corticosterone [41]. Concentrations of 30 nM and 100 nM of dexamethasone significantly increased HMGB1 release after 24 h treatment (Figure 2A). A lower concentration of dexamethasone tended to increase extracellular HMGB1. Concentrations of up to 10 μM dexamethasone were not cytotoxic, indicating that HMGB1 released with either 30 nM or 100 nM of dexamethasone was not a result of nonspecific cytotoxicity, necrosis, or apoptosis (Appendix A). Significant extracellular HMGB1 was observed 24 h after treatment with 100 nM dexamethasone, which tended to be elevated for at least 72 h (Figure 2A).

### 3.3. Effects of Corticosterone on Intracellular Localization of HMGB1 in Rat Primary Cultured Cortical Astrocytes

HMGB1 translocates from the nucleus into the cytoplasm before release into the extracellular space [42]. HMGB1 expression was localized mainly in the nuclear compartment in vehicle-treated astrocytes (Figure 3A). Twenty-four hours after treatment with 1 µM corticosterone, HMGB1 clearly translocated to the cytosol in astrocytes (Figure 3A). The percentage of cells with HMGB1 staining in the nuclei decreased after corticosterone treatment, compared with vehicle treatment (Figure 3B). The percentage of cells with cytoplasmic HMGB1 increased after corticosterone treatment, compared with vehicle treatment (Figure 3C).

### 3.4. Effects of Corticosterone on HMGB1 mRNA and Intracellular HMGB1 Protein Expression in Rat Primary Cultured Cortical Astrocytes

It is possible that corticosterone, which induces gene transcription, upregulated HMGB1 mRNA expression. Concentrations up to 10 µM of corticosterone for 24 h did not affect HMGB1 mRNA and total intracellular HMGB1 protein expression (Figure 4A,B). No significant changes in HMGB1 mRNA and intracellular HMGB1 protein expression were observed during the entire 24 h incubation period with 1 µM corticosterone (Figure 4A,B).

### 3.5. Effects of Glucocorticoid Receptor Antagonist on Corticosterone or Dexamethasone-Induced HMGB1 Release in Rat Primary Cultured Cortical Astrocytes

Involvement of the glucocorticoid receptor in the corticosterone-induced release of HMGB1 from astrocytes was confirmed with mifepristone, a glucocorticoid receptor antagonist. Pretreatment with 0.5 µM mifepristone significantly reduced extracellular HMGB1 following both 1 µM corticosterone and 100 nM dexamethasone treatment (Figure 5A,B). Treatment with antagonist alone, without either corticosterone or dexamethasone, did not affect extracellular HMGB1 (Figure 5A,B).

### 3.6. The Effects of Pannexin-1 Inhibitors and P2X7 Antagonists on Corticosterone-Induced HMGB1 Release in Rat Primary Cultured Cortical Astrocytes

Corticosterone treatment increased adenosine triphosphate (ATP) release through pannexin-1 hemichannel in primary cultured astrocytes [43]. Pannexin-1 hemichannel-mediated ATP efflux, and then extracellular ATP binding to P2X7 receptors mediate lipopolysaccharide (LPS)-induced HMGB1 release in macrophage [31]. Thus, it is possible that pannexin-1 and P2X7 receptors are involved in corticosterone-induced HMGB1 release in primary cultured cortical astrocytes. Cells were treated overnight with ^10^Panx, a selective pannexin-1 inhibitor [30], and then treated with corticosterone. For other antagonists, cells were treated for 30 min before the start of the 24 h corticosterone treatment. Pretreatment with 10 µM carbenoxolone, an inhibitor of both connexins and pannexins [44], and 200 µM ^10^Panx, significantly decreased HMGB1 release induced by 1 µM corticosterone treatment (Figure 6A,B). Pretreatment with 5 µM A804598, and 5 µM A438079, P2X7 antagonists, significantly decreased corticosterone-induced HMGB1 release (Figure 6C). Treatment with either inhibitors or antagonists alone, without corticosterone treatment, did not affect HMGB1 levels in media.

## 4. Discussion

The current study demonstrated that corticosterone, a major stress hormone in rodents, directly stimulates astrocytes through the glucocorticoid receptor and triggers cytoplasmic translocation and extracellular release of nuclear HMGB1. The release of HMGB1 involves pannexin-1 and P2X7 signaling. Once released into the extracellular space, HMGB1 is found in a number of processes including inflammation through its receptors such as TLR4 and RAGE [45,46]. Thus, glucocorticoid-mediated HMGB1 release from astrocytes could be involved in neuroinflammation, which, in turn, mediates stress and, by extension, the pathophysiology of MDD.

It was previously thought that the release of HMGB1 occurred via passive nonspecific leakage from necrotic cells [47,48]. However, HMGB1 can be actively secreted via specific signaling mechanisms as shown in previous studies and in the current study [49,50]. The concentrations of corticosterone and dexamethasone used in the current study did not induce cytotoxicity or the loss of membrane integrity, indicating that HMGB1 found in the media was likely extruded from astrocytes via an active process. HMGB1 has multiple intracellular functions, depending on its cellular distribution, and extracellular functions following release [51]. Location of HMGB1 is intricately linked with its function and is regulated by a series of posttranslational modifications. HMGB1 undergoes extensive post-translational modifications, in particular acetylation and oxidation which modulate its function [52]. HMGB1 actively secreted by immune cells is acetylated in both thiol and disulfide isoforms [53]. The response of CNS cells, particularly astrocytes, has not been extensively studied. In addition to HMGB1 itself, identifying the isoforms of HMGB1 released from astrocyte by corticosterone will be important in the further elaboration of HMGB1’s role in normal and pathological states.

Corticosterone is known to interact with the glucocorticoid receptor and the mineralocorticoid receptor, both of which are expressed in cultured astrocytes [54]. In the current study, blocking the glucocorticoid receptor significantly attenuated corticosterone- and dexamethasone-induced HMGB1 release. Dexamethasone has little affinity for the mineralocorticoid receptor [55]. Thus, the current study demonstrated that glucocorticoid-mediated HMGB1 release from astrocytes is mediated through glucocorticoid receptors.

The source of extracellular HMGB1 observed in the current study likely existed in nuclear HMGB1 rather than newly synthesized HMGB1. The current study demonstrated that corticosterone treatment did not increase HMGB1 mRNA and total HMGB1 protein expression in primary cultured astrocytes. Immunocytochemistry of corticosterone-treated astrocytes showed translocation of HMGB1 from the nucleus to the cytosol—there were fewer astrocytes demonstrating nuclear HMGB1 staining following corticosterone treatment compared to vehicle-treated astrocytes and, conversely, there were more astrocytes with cytosolic HMGB1 staining following corticosterone treatment compared to vehicle-treated astrocytes.

Active secretion of HMGB1 has been observed in various immune cells and non-immune cells, including monocytes, macrophages [56], endothelial cells [57], fibroblasts [58], and brain glial cells [24]. HMGB1 is not released via the canonical endoplasmic reticulum/Golgi pathway. The mechanism by which active secretion of HMGB1 is complex, varies among cell types and according to the stimulus. A previous study showed that IL-1β-stimulated astrocytes upregulated and released HMGB1 via specific signaling pathways involving extracellular signal-regulated kinase (ERK) and the nuclear protein exporter, chromosome region maintenance 1 (CRM1) [25]. However, U0126, a MAPK/ERK kinase inhibitor, did not have any effect on corticosterone-induced HMGB1 release from astrocyte (data not shown). A selective CRM1 inhibitor, Leptomycin B, could not be used because of its cytotoxicity. Other cellular mechanisms that could be involved in astrocytic secretion of HMGB1 remain to be uncovered.

A previous study demonstrated that corticosterone increased serum- and glucocorticoid-inducible kinase-1 (SGK-1) expression through the glucocorticoid receptor, and SGK-1 enhanced ATP release through pannexin-1 hemichannel in primary cultured astrocytes [43]. Thus, it is possible that SGK-1 and pannexin-1 are involved in the corticosterone-induced release of HMGB1 in primary cultured cortical astrocytes. In the current study, SGK-1 mRNA expression is increased 3 h after corticosterone treatment and, furthermore, SGK-1 mRNA expression is sustained, being increased 24 h after corticosterone treatment (data not shown). Based on this, we speculate that ATP release through astrocytic hemichannels is sustained long after corticosterone treatment. However, it is difficult to further elucidate the involvement of SGK-1 because GSK650394, a selective SGK-1 inhibitor, is cytotoxic. Future studies involving, for example, short interfering RNAs specific to SGK-1, could further clarify the mechanism of astrocytic glucocorticoid-induced HMGB1 release. Glucocorticoids open astrocytic pannexin-1 channels, thereby releasing ATP [59]. The current findings also showed that blocking connexins and pannexins, and specifically pannexin-1, led to significantly decreased HMGB1 release, demonstrating that pannexin-1 is involved in corticosterone-induced HMGB1 release in primary cultured cortical astrocytes.

Glucocorticoid enhance ATP release, which leads to activation of purinergic receptors expressed on astrocytes themselves [60,61]. Pannexin-1-mediated ATP efflux and subsequent P2X7 receptor activation have been suggested as mediating lipopolysaccharide (LPS)-induced HMGB1 release from macrophage [31]. In the current study, blocking astrocytic P2X7 receptors led to significantly decreased corticosterone-induced HMGB1 release. Interestingly, a previous study showed that extracellular ATP is robustly increased in the hippocampus during acute immobilization stress and chronic unpredictable stress, which led to increased cytokine release, due to nucleotide-binding, leucine-rich repeat, pyrin domain containing 3 (NLRP3) inflammasome activation, and increased depressive-like behaviors, which were attenuated with block of the P2X7 receptor [62,63]. Formation of the NLRP3 inflammasome and activation of caspase-1 downstream signaling of the P2X7 receptor, have also been implicated as regulatory mechanisms of extracellular HMGB1 release [64]. Dexamethasone exposure increased the expression of NLRP3 inflammasome pathway in a rat microglial cell line [65]. Thus, P2X7 receptor-mediated NLRP3 inflammasome activation could be involved in corticosterone-induced HMGB1 release from astrocytes.

Glucocorticoids have been universally regarded as anti-inflammatory but a considerable number of studies now demonstrate that under specific conditions, glucocorticoids are capable of potentiating neuroinflammatory processes (i.e., priming) [66]. In light of current findings, it is proposed that stress-induced increases in glucocorticoids activates glucocorticoid receptors and subsequently increases HMGB1 release from CNS astrocyte. Extracellular HMGB1 interact with TLR2, TLR4, and RAGE which express in astrocyte and microglia [67,68] to induce innate immunity activation [69]. Astrocyte-derived HMGB1 could then be involved in neuroinflammatory processes which, in turn, mediate psychiatric disorders such as MDD. The current study focused on HMGB1 found in astrocytes, which are the most abundant cell type in the CNS. It is possible that other CNS cell types produce HMGB1 in response to stress, such as neurons and microglia. For example, chronic unpredictable stress causes robust up-regulation of HMGB1 mRNA in hippocampal microglia [18]. There are no reports of HMGB1 released from neurons under stressful conditions.

Taken together, the current study demonstrates that corticosterone acts directly on astrocytes and evokes the release of HMGB1 through a pannexin-1 and P2X7 receptor signaling cascade. Extracellular HMGB1 could then contribute to the development or maintenance, or both, of neuroinflammation that mediates psychiatric disorders such as MDD. Thus, HMGB1, and its upstream and downstream signaling molecules, may serve as therapeutic targets for the prevention or treatment of a number of psychiatric disorders.

## Figures and Tables

**Figure 1 cells-09-01068-f001:**
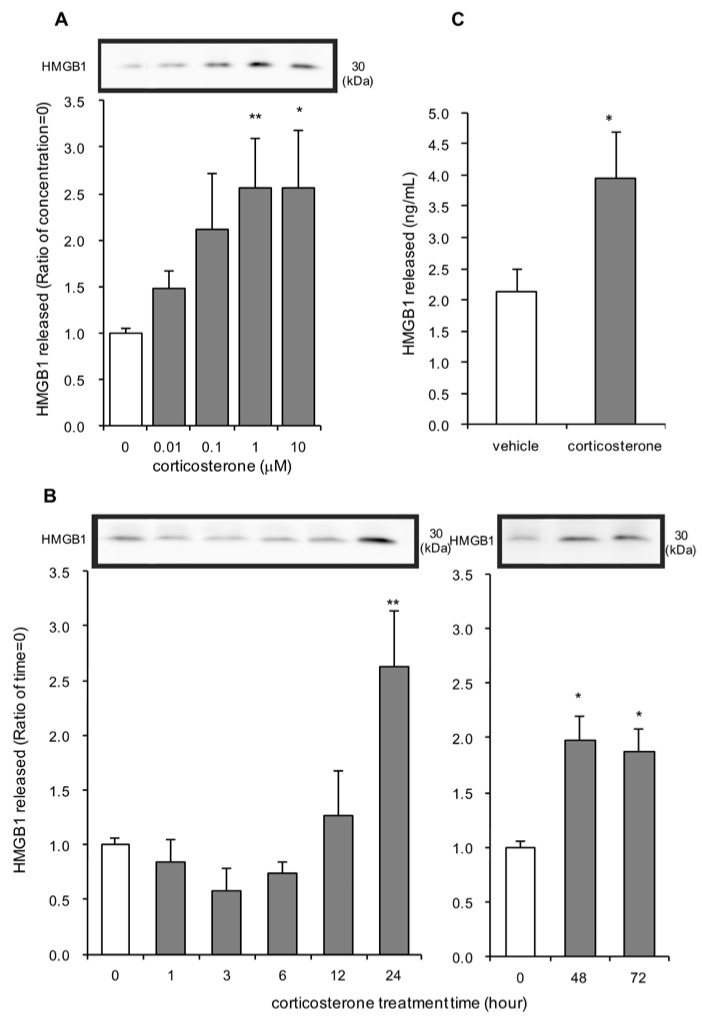
The effect of corticosterone on high-mobility group box-1 (HMGB1) release in primary cultured cortical astrocytes. (**A**) Increasing concentrations of corticosterone induced HMGB1 release from astrocytes. Cells were treated with the indicated concentrations of corticosterone for 24 h, and the amount of HMGB1 in the media was analyzed by Western blotting. Representative blots are shown (HMGB1: 30 kDa). The data are expressed as the mean ± SEM. * *p* < 0.05, ** *p* < 0.01 vs. basal (concentration “0”) (Dunnett’s test; *n* = 7–15). (**B**) Astrocytes were treated with 1 μM corticosterone for the indicated periods of time and the amount of HMGB1 in the media was analyzed by Western blotting. Representative blots are shown (HMGB1: 30 kDa). The data are expressed as the mean ± SEM. * *p* < 0.05, ** *p* < 0.01 vs. basal (time “0”) (Dunnett’s test; *n* = 3–15). (**C**) Astrocytes were treated with 1 μM corticosterone for 24 h, and media concentrations of HMGB1 were analyzed by ELISA. The data are expressed as the mean ± SEM. * *p* < 0.05 vs. vehicle (*t*-test; *n* = 5).

**Figure 2 cells-09-01068-f002:**
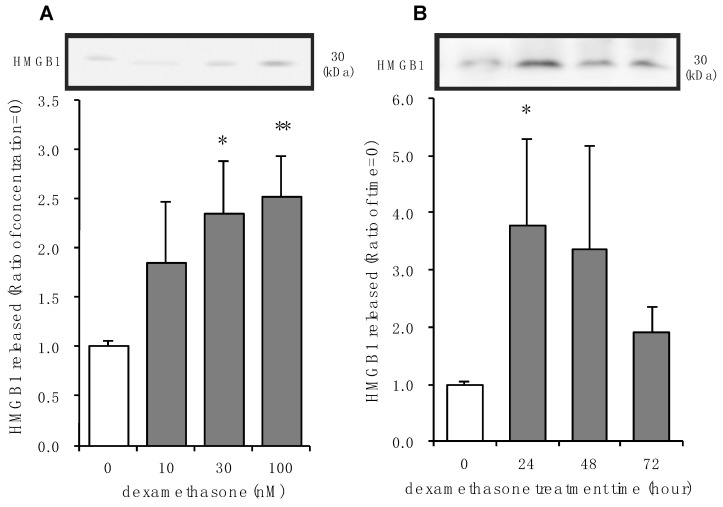
The effect of dexamethasone on HMGB1 release in primary cultured cortical astrocytes. (**A**) Increasing concentrations of dexamethasone induced HMGB1 release from astrocytes. Cells were treated with the indicated concentrations of dexamethasone for 24 h, and the amount of HMGB1 in the media was analyzed by Western blotting. Representative blots are shown (HMGB1: 30 kDa). The data are expressed as the mean ± SEM. * *p* < 0.05, ** *p* < 0.01 vs. basal (concentration “0”) (Dunnett’s test; *n* = 4–10). (**B**) Astrocytes were treated with 100 nM dexamethasone for the indicated periods of time and the amount of HMGB1 in the media was analyzed by Western blotting. Representative blots are shown (HMGB1: 30 kDa). The data are expressed as the mean ± SEM. * *p* < 0.05 vs. basal (time “0”) (Dunnett’s test; *n* = 5–12).

**Figure 3 cells-09-01068-f003:**
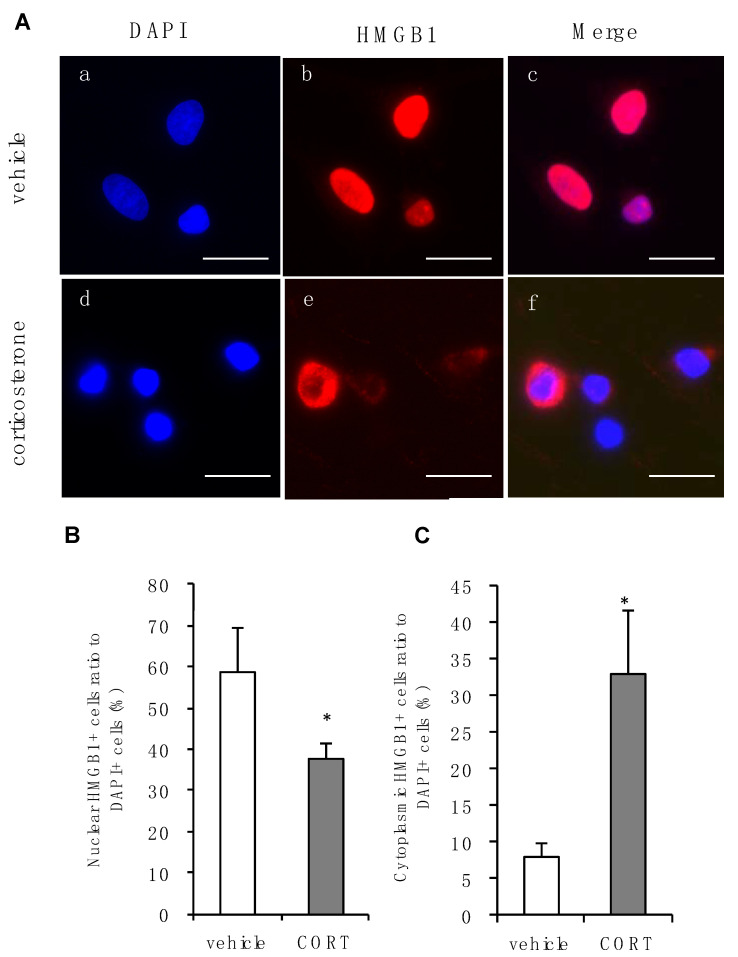
Effects of corticosterone on intracellular localization of HMGB1 in primary cultured cortical astrocytes. (**A**) Immunocytochemistry of HMGB1 in rat primary cultured cortical astrocytes. Corticosterone-treated cells (CORT; 1 μM, 24 h) exhibited subcellular HMGB1 expression compared with vehicle. Images a, b and c, or d, e and f, are from the same fields, respectively. Scale bar = 20 μm. (**B**) The percentage of nuclear staining of HMGB1 to total DAPI positive cells decreased after corticosterone treatment. The data are expressed as the mean ± SEM. * *p* < 0.05 vs. vehicle (*t*-test; *n* = 3–4). (**C**) The percentage of cytoplasmic staining of HMGB1 to total DAPI positive cells increased after corticosterone treatment. The data are expressed as the mean ± SEM. * *p* < 0.05 vs. vehicle (*t*-test; *n* = 3–4).

**Figure 4 cells-09-01068-f004:**
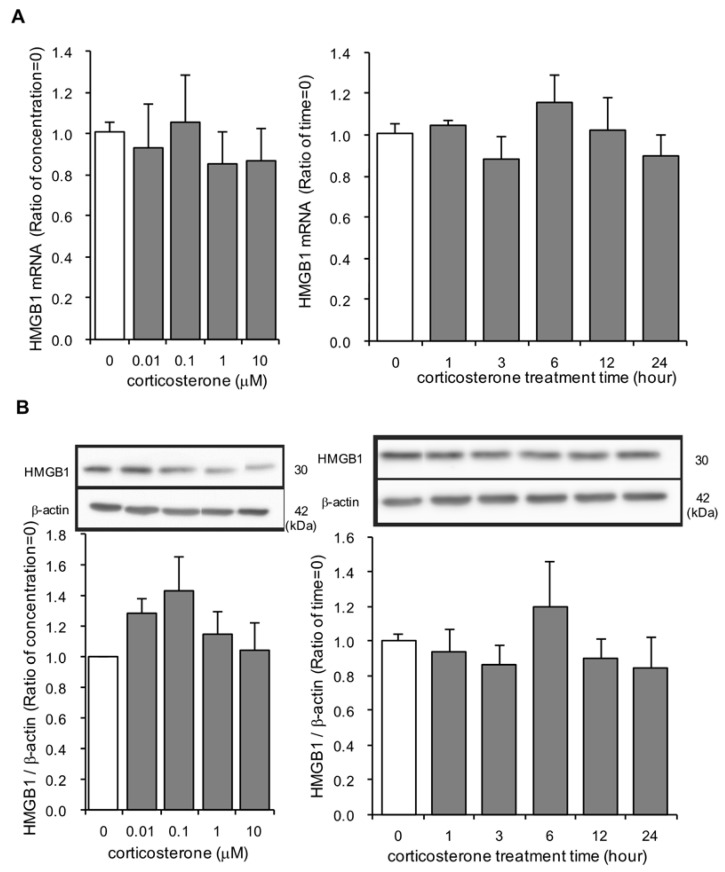
The effect of corticosterone on HMGB1 mRNA and intracellular HMGB1 protein expression in primary cultured cortical astrocytes. (**A**) Astrocytes were treated with 1 μM corticosterone for the indicated periods of time or the indicated concentrations of corticosterone for 24 h. HMGB1 mRNA expression (ratio of HMGB1 mRNA: GAPDH mRNA) was quantified by real time PCR. The data are expressed as the mean ± SEM. (*n* = 3–5). (**B**) Astrocytes were treated with 1 μM corticosterone for the indicated periods of time or the indicated concentrations of corticosterone for 24 h. HMGB1 protein was quantified by Western blotting, and representative blots are shown (HMGB1: 30 kDa, β-actin: 42 kDa). The data are expressed as the mean ± SEM. (*n* = 3–5).

**Figure 5 cells-09-01068-f005:**
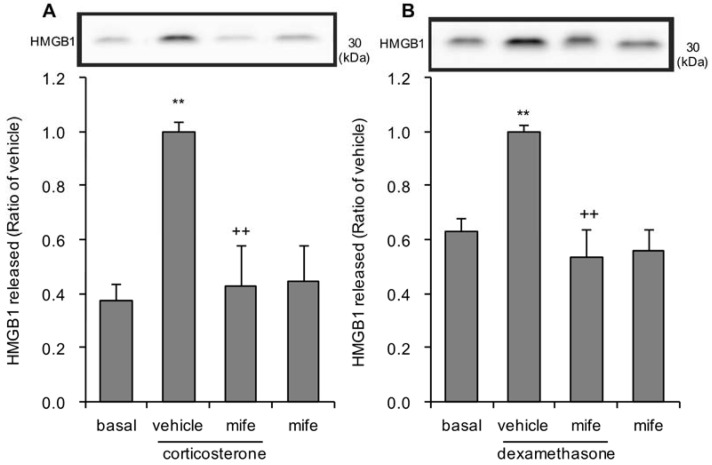
Effects of glucocorticoid receptor antagonist on corticosterone or dexamethasone-induced HMGB1 release in primary cultured cortical astrocytes. (**A**) Astrocytes were pretreated with 0.5 μM mifepristone (mife) for 30 min, and subsequently treated with 1 μM corticosterone for 24 h. The amount of HMGB1 in the media was quantified by Western blotting. Representative blots are shown (HMGB1: 30 kDa). The data are expressed as the mean ± S.E.M. ** *p* < 0.01 vs. basal and ^++^
*p* < 0.01 vs. corticosterone alone. (Tukey’s HSD test; *n* = 6–10). (**B**) Astrocytes were pretreated with 0.5 μM mifepristone (mife) for 30 min, and subsequently treated with 100 nM dexamethasone for 24 h. The amount of HMGB1 in the media was quantified by Western blotting. Representative blots are shown (HMGB1: 30 kDa). The data are expressed as the mean ± S.E.M. ** *p* < 0.01 vs. basal and ^++^
*p* < 0.01 vs. dexamethasone alone. (Tukey’s HSD test; *n* = 5–9).

**Figure 6 cells-09-01068-f006:**
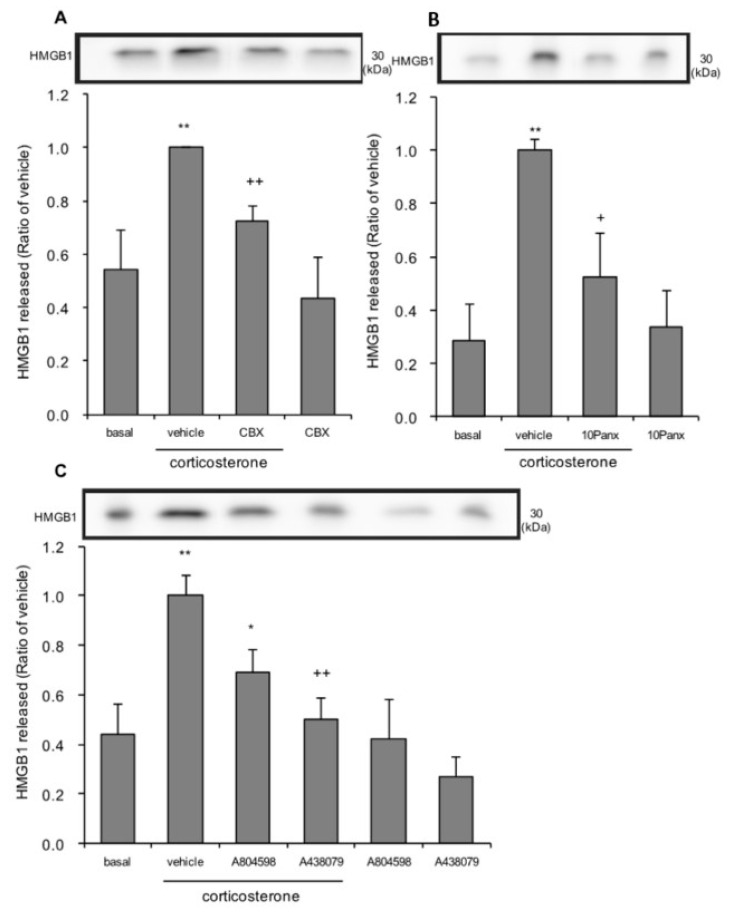
The effects of pannexin-1 inhibitors and P2X7 antagonists on corticosterone-induced HMGB1 release in primary cultured cortical astrocytes. (**A**) Astrocytes were pretreated with 10 μM carbenoxolone (CBX) for 30 min, and subsequently treated with 1 μM corticosterone for 24 h. The amount of HMGB1 in the media was quantified by Western blotting. Representative blots are shown (HMGB1: 30 kDa). The data are expressed as the mean ± SEM. ** *p* < 0.01 vs. basal and ^++^
*p* < 0.01 vs. corticosterone alone. (Tukey’s HSD test; *n* = 6–7). (**B**) Astrocytes were pretreated with 200 μM ^10^Panx overnight, and subsequently treated with 1 μM corticosterone for 24 h. The amount of HMGB1 in the media was quantified by Western blotting. Representative blots are shown (HMGB1: 30 kDa). The data are expressed as the mean ± SEM. ** *p* < 0.01 vs. basal and ^+^
*p* < 0.05 vs. corticosterone alone. (Tukey’s HSD test; *n* = 4–6). (**C**) Astrocytes were pretreated with either 5 μM A804598 or 5 μM A438079 for 30 min, and subsequently treated with 1 μM corticosterone for 24 h. The amount of HMGB1 in the media was quantified by Western blotting. Representative blots are shown (HMGB1: 30 kDa). The data are expressed as the mean ± SEM. ** *p* < 0.01 vs. basal and * *p* < 0.05, ^++^
*p* < 0.01 vs. corticosterone alone. (Tukey’s HSD test; *n* = 3–10).

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
