# Peer review of "Corticosterone Induces HMGB1 Release in Primary Cultured Rat Cortical Astrocytes: Involvement of Pannexin-1 and P2X7 Receptor-Dependent Mechanisms"

_cells, 2020, doi:10.3390/cells9051068_

Round 1
Reviewer 1 Report
HMGB1 has been implicated as mediating several various sterile inflammatory processes. In this paper the authors have investigated its potential role in major depressive disorder (MDD), an area without much cellular/molecular level research available. They found that stress-associated glucocorticoids (GCs) may be responsible for releasing the HMGB1 from astrocytes. Further, they show that inhibition of 2 receptors, pennexin-1 and P2X7, decreased GCs and levels of HMGB1. This work is fairly novel and interesting, but suffers from the superficial relation back to the known literature on the function of HMGB1. The authors need to expand on the relevance of their study in the context of what has already been published and address the following major concerns.
Comments:
1. The logic found in Lines 67-69 is unclear. How are astrocytes considered the main source of secreted HMGB1 just because they contain more intracellular HMGB1 following stress? It is known that nuclear HMGB1 translocates to the cytoplasmic vesicles for secretion to extracellular spaces, but not that intracellular HMGB1 levels are increased as a result of stress. The authors should expand on this idea and provide references. This is especially important in light of their data shown in Figure 4.
2. Line 209, the authors state that the astrocytic release of HMGB1 is not due to necrosis or nonspecific cytotoxicity, but is it due to apoptosis of the astrocytes? There have been reports that different oxidative forms of HMGB1 are released related to different forms of cell death and the authors should address this.
3. Figure 3 shows astrocytes translocating HMGB1 from their nucleus to their cytoplasmic compartment. The images are not high resolution enough to determine if the cytoplasmic HMGB1 is restricted within LAMP-1+ vesicles, indicating their future active release rather than passive release due to cell death. The authors should comment on whether they observed the cytoplasmic HMGB1 to be vesicle-restricted or diffuse in astrocytes to determine if the astrocytes are actively secreting it or if it simply reflects their loss of membrane integrity.
4. Similar to comment 2, the authors should ensure the pennexin-1 inhibitor and P2X7 antagonists, etc. did not induce any kind of cell death programs including apoptosis which could have affected the type of HMGB1 that was released and therefore its downstream effects.
5. Similar to comment 1, the paragraph in Lines 308-312 does not adequately describe the current literature regarding HMGB1. A better literature review is needed describing HMGB1 localization and its varied effects.
6. Lines 44-50 describe the various receptors that HMGB1 can signal through but this is not revisited within the discussion section. The authors should comment on their data and the possibility for downstream signaling following astrocyte release.
7. Along these same lines, can the authors please describe which HMGB1 receptors astrocytes may express. If they are unable to test for this in their own cultured system then at least they should report what is known in the literature. It seems more likely that microglia play the role of actively secreting HMGB1 and astrocytes are simply reacting to the stress of GC treatment since macrophages/monocytes outside of the brain have been shown to perform this role rather than parenchymal cells. The discussion of the potential role of microglia on Lines 327-329 is a good start, and discussion regarding the HMGB1-specific receptors that they express should be expanded to address this.
8. The authors should also expand their discussion of the potential use of GCs in the treatment of MDD. It is unknown from the manuscript itself whether the authors are suggesting GCs should be used as a treatment or that they are used for other ailments that individuals with MDD may have and they may affect the MDD.
Author Response
Point-by-point reply to Reviewer 1
1) Reviewer 1’s comment
The logic found in Lines 67-69 is unclear. How are astrocytes considered the main source of secreted HMGB1 just because they contain more intracellular HMGB1 following stress? It is known that nuclear HMGB1 translocates to the cytoplasmic vesicles for secretion to extracellular spaces, but not that intracellular HMGB1 levels are increased as a result of stress. The authors should expand on this idea and provide references. This is especially important in light of their data shown in Figure 4.
Response:
In the current study, we focused on astrocytes because astrocytes are the most abundant cell type in the CNS. A previous study showed that stress increased HMGB1 in the brain tissue as measured in brain lysate by western blotting (Cheng, et al., 2016, Brain Behav Immun; Wang et al., 2018, Brain Behav Immun), however the mechanism of this increase and the cell which actually showed increased HMGB1 is largely unknown. Brain astrocytes express HMGB1 in vivo (Kim et al., 2008, J Neurosci Res), and release HMGB1 in response to inflammogens and second messenger activators in vitro (Passalacqua et al., 1997, FEBS Lett; Hayakawa et al., 2010, Glia).
The Introduction and Discussion section has been edited:
Revised Introduction
Lines 60-61. “However, the mechanism of stress-induced increase of HMGB1 and the cell which actually showed increased HMGB1 is largely unknown.”
Lines 69-70. “Thus, astrocyte be the one of source of extracellular HMGB1 under conditions of stress.”
Revised Discussion
Line 384-388. “The current study focused on HMGB1 found in astrocytes, which are the most abundant cell type in the CNS. It is possible that other CNS cell types produce HMGB1 in response to stress, such as neurons and microglia. For example, chronic unpredictable stress causes robust up-regulation of HMGB1 mRNA in hippocampal microglia [18]. There are no reports of HMGB1 released from neurons under stressful conditions.”
2) Reviewer 1’s comment
Line 209, the authors state that the astrocytic release of HMGB1 is not due to necrosis or nonspecific cytotoxicity, but is it due to apoptosis of the astrocytes? There have been reports that different oxidative forms of HMGB1 are released related to different forms of cell death and the authors should address this.
Response:
There are two patterns of HMGB1 release, active release and passive release. Active release of HMGB1 occurs through activation of a defined intracellular signaling pathway. Passive release of HMGB1 can occur following cell death. At very high concentrations (> 10 μM), corticosterone could lead to apoptosis and HMGB1 release (Bell et al., 2006, Am J Phisol Cell Physiol). Generally, however, apoptosis does not lead to HMGB1 release (Scaffidi et al., 2002, Nature). The concentration of corticosterone and dexamethasone used in the current study did not induce apoptosis. Thus, HMGB1 release in the current study was not due to diffusion due to cell apoptosis. HMGB1 release in the current study was mechanism-based rather than due to cell death.
We have also conducted MTT assay as well as LDH assay as shown in Supplemental Fig. 1. Concentrations of corticosterone and dexamethasone up to 10 μM over a period of 24 h did not have any effect on cell viability in MTT assay. Data of MTT assay have added to Supplemental Fig. 2.
The wording of line 209 has been changed to clearly state that release of HMGB-1 was mechanism-mediated rather than diffusion via a pathological process.
Revised Results
Lines 204-207. “Concentrations of up to 10 μM dexamethasone were not cytotoxic, indicating that HMGB1 released with either 30 nM or 100 nM of dexamethasone was not a result of nonspecific cytotoxicity, necrosis, or apoptosis (Supplemental Fig 1B, Supplemental Fig 2B).”
3) Reviewer 1’s comment
Figure 3 shows astrocytes translocating HMGB1 from their nucleus to their cytoplasmic compartment. The images are not high resolution enough to determine if the cytoplasmic HMGB1 is restricted within LAMP-1+ vesicles, indicating their future active release rather than passive release due to cell death. The authors should comment on whether they observed the cytoplasmic HMGB1 to be vesicle-restricted or diffuse in astrocytes to determine if the astrocytes are actively secreting it or if it simply reflects their loss of membrane integrity.
Response:
HMGB1 is secreted by monocytes via a non-classical, vesicle-mediated secretory pathway (Gardella et al., 2002, EMBO Rep). With respect to astrocytes releasing HMGB1, however, there are no reports.
Based on the experimental conditions of the current study, HMGB1 release in the current study was mechanism-based rather than due to cell death. While it is possible that HMGB1 can be found within vesicles, it is also possible that HMGB1 was diffuse within the cytoplasm. Because we do not have higher resolution images, we cannot be sure if HMGB1 was restricted to vesicles. We will speculate, though, that since HMGB1 was released in a controlled manner from living cells, it is possible that HMGB1 was in vesicles.
Revised Discussion
Lines 311-314. “The concentrations of corticosterone and dexamethasone used in the current study did not induce cytotoxicity or the loss of membrane integrity, indicating that HMGB1 found in the media was likely extruded from astrocytes via an active process.”
4) Reviewer 1’s comment
Similar to comment 2, the authors should ensure the pennexin-1 inhibitor and P2X7 antagonists, etc. did not induce any kind of cell death programs including apoptosis which could have affected the type of HMGB1 that was released and therefore its downstream effects.
Response:
The concentrations of carbenoxolone, 10Panx, and antagonists (mifepristone, A804598, and A438079) alone, without corticosterone treatment, did not induced cell death. Other studies demonstrated a lack of cell death at the concentrations of compounds used in the current study (Woehrle et al., 2010, J Leukoc Biol; Li et al., 2013, Mol Med; Unemura et al., 2012, J Pharmacol Sci; Liu et al., 2020, Am J Transl Res; Pena-Altamira et al., 2018, Neurochem Int).
5) Reviewer 1’s comment
Similar to comment 1, the paragraph in Lines 308-312 does not adequately describe the current literature regarding HMGB1. A better literature review is needed describing HMGB1 localization and its varied effects.
Response:
The discussion concerning HMGB1 is restricted to relevance to the findings in the current study and a general overall relevance to previous findings, rather than a long exposition on HMGB1 in general. In particular, we would like to focus on “astrocytes”.
Revised Discussion
Lines 314-322. “HMGB1 has multiple intracellular functions, depending on its cellular distribution, and extracellular functions following release [51]. Location of HMGB1 is intricately linked with its function and is regulated by a series of posttranslational modifications. HMGB1 undergoes extensive post-translational modifications, in particular acetylation and oxidation which modulate its function [52]. HMGB1 actively secreted by immune cells is acetylated and both thiol- and disulfide isoforms [53]. The response of CNS cells, particularly astrocytes, has not been extensively studied. In addition to HMGB1 itself, identifying the isoforms of HMGB1 released from astrocyte by corticosterone will be important in further elaboration HMGB1’s role in normal and pathological states.”
6) Reviewer 1’s comment
Lines 44-50 describe the various receptors that HMGB1 can signal through but this is not revisited within the discussion section. The authors should comment on their data and the possibility for downstream signaling following astrocyte release.
Response:
The focus of the current study was not HMGB1 receptor subtypes as implied by the Reviewer’s comment but astrocytic release of HMGB1:
Revised Discussion
Lines 305-306 “Once released into the extracellular space, HMGB1 is found in a number of processes including inflammation through its receptors such as TLR4 and RAGE [45,46].”
7) Reviewer 1’s comment
Along these same lines, can the authors please describe which HMGB1 receptors astrocytes may express. If they are unable to test for this in their own cultured system then at least they should report what is known in the literature. It seems more likely that microglia play the role of actively secreting HMGB1 and astrocytes are simply reacting to the stress of GC treatment since macrophages/monocytes outside of the brain have been shown to perform this role rather than parenchymal cells. The discussion of the potential role of microglia on Lines 327-329 is a good start, and discussion regarding the HMGB1-specific receptors that they express should be expanded to address this.
Response:
There is a report showing that astrocytes express HMGB1 receptor such as TLR2, TLR4, and RAGE (Qiu et al., 2010, Stroke). It is controversial about roles of released HMGB1 from astrocytes under conditions of stress. One possibility is activation of microglia. In fact, microglia also express HMGB1 receptors, including TLR4 and RAGE, and HMGB1 activates microglia (Rosciszewski et al., 2019, Front Cell Neurosci). The previous study showed that increased HMGB1 might induce Iba1 expression in frontal cortex of mice following peripheral nerve injury, which present depressive-like behavior (Hisaoka-Nakashima et al., 2019, Prog Neuropsychopharmacol Biol Psychiatry). Then, it is necessary to elaborate whether released HMGB1 from astrocytes could directly activate microglia located closely and mediate their function.
Revised Discussion
Lines 381-383. “Extracellular HMGB1 interact with TLR2, TLR4, and RAGE which express in astrocyte and microglia [67,68] to induce innate immunity activation [69].”
8) Reviewer 1’s comment
The authors should also expand their discussion of the potential use of GCs in the treatment of MDD. It is unknown from the manuscript itself whether the authors are suggesting GCs should be used as a treatment or that they are used for other ailments that individuals with MDD may have and they may affect the MDD.
Response:
The Reviewer probably knows that glucocorticoids (GCs) are not routinely used in the treatment of MDD. Our findings suggest that perhaps GCs antagonists could be used for the treatment of MDD. There are reports that GCs antagonists have an antidepressants effect on animal model (Zhang et al., 2018, Behav Brain Res). Increased corticosterone could be used as a biomarker of MDD. In fact, a number of stress factors are elevated, any of which could lead to HMGB1 release or translocation to the cytoplasm, as shown in preclinical studies. Clinical studies will be needed to confirm these observations.
Reviewer 2 Report
The manuscript by Hisaoka-Nakashima et al. studies whether glucocorticoids, a major signal during chronic stress, induce HMGB1 release from astrocytes, which in turn has been implicated in inflammatory processes secondary to stress. The article is concise and well-written. The hypothesis is clearly stated and consistent with previously published data. The methodology appears correct, although I do have several specific comments regarding methodological aspects (see below). The conclusions are in agreement with the data. My main suggestions have to do with the discussion of the data, as well as improving the explanation of methodology and presentation of data. I also have some minor corrections to the text.
Specific comments:
- Lines 91-95: this paragraph is a justification of the experiments examining the role of pannexin-1 and P2X7 receptors in this system. It seems to be out of place in Material & Methods. It should probably be inserted before line 277 to introduce the corresponding experiments.
- Related to the previous point, the big question remains how to explain the proposed ATP efflux upon glucocorticoid treatment, particularly given the slow kinetics of the effect. In the Discussion (lines 342-343) the authors state that the GR target gene SGK1 could be involved, since there is a previous reference indicating that SGK1 induces ATP release from astrocytes. The authors mention that SGK1 is induced by corticosterone in their system (data not shown). However, SGK1 is normally an early glucocorticoid-induced gene. The kinetics of this induction in the experiments described in the manuscript is a potentially important piece of information to assess whether this could be the pathway increasing the proposed (but not proven) ATP release.
- When were Tukey´s and Dunnett´s tests used and what was the criteria to use one or the other?
- Carbenoxolone is a notoriously unspecific drug. In particular, it inhibits 11-beta-HSD2, an enzyme involved in converting glucocorticoids to inactive metabolites. This is particularly important in cells that co-express GR and MR Do astrocytes express HSD2? If so, could carbenoxolone alter the response to glucocorticoids?
- How do you normalize the western blots obtained from supernatant (Fig.1, Fig.2)? Data from whole-cell lysates could be used cells (Fig.4), but it is unclear if this was the way the experiments were performed.
- Pictures in Fig.3A do not seem to be representative of the data described in B. It would be nice to see a field with cells where HMGB1 is still nuclear and others with cytosolic localization.
- Regarding Fig.5 and Fig.6, why not follow the criteria from Fig.1, Fig.2 and Fig.4 and normalize data to basal expression of HMGB1?
- Line 87: correct “carbenoxolne”, it should be “carbenoxolone”
- Line 188: “confirm” should be “confirmed”
Author Response
Point-by-point reply to Reviewer 2
1) Reviewer 2’s comment
Lines 91-95: this paragraph is a justification of the experiments examining the role of pannexin-1 and P2X7 receptors in this system. It seems to be out of place in Material & Methods. It should probably be inserted before line 277 to introduce the corresponding experiments.
Response:
The paragraph (Lines 91-95) has been moved to Lines 274-278.
2) Reviewer 2’s comment
Related to the previous point, the big question remains how to explain the proposed ATP efflux upon glucocorticoid treatment, particularly given the slow kinetics of the effect. In the Discussion (lines 342-343) the authors state that the GR target gene SGK1 could be involved, since there is a previous reference indicating that SGK1 induces ATP release from astrocytes. The authors mention that SGK1 is induced by corticosterone in their system (data not shown). However, SGK1 is normally an early glucocorticoid-induced gene. The kinetics of this induction in the experiments described in the manuscript is a potentially important piece of information to assess whether this could be the pathway increasing the proposed (but not proven) ATP release.
Response:
SGK-1 appears to enhance ATP release through the opening of pannexin-1 hemichannels. While the current study did not measure ATP release following corticosterone treatment, we hypothesize that the ATP release is increased and sustained due to increased expression of SGK-1. A previous study (Koyanagi et al., 2016, Nature Communications) demonstrated that corticosterone treatment in astrocytes increased SGK-1 mRNA after 3 hr of treatment and ATP release was increased 4 hr after treatment. Based on Koyanagi et al., it is unknown whether the effect of corticosterone on ATP release is transient or sustained. We found (data not shown) that SGK-1 mRNA expression is increased 3 hr after corticosterone treatment and, furthermore, SGK-1 mRNA expression is sustained, being increased 24 hr after corticosterone treatment. Based on this, we speculate that ATP release through astrocytic hemichannels is sustained long after corticosterone treatment. Thus, corticosterone-mediated HMGB1 release from astrocytes, mediated in part by ATP, appears 3 hrs after corticosterone treatment and could be sustained for at least 24 hrs after treatment.
Revised Discussion
Lines 352-355. “In the current study, SGK-1 mRNA expression is increased 3 hr after corticosterone treatment and, furthermore, SGK-1 mRNA expression is sustained, being increased 24 hr after corticosterone treatment (data not shown). Based on this, we speculate that ATP release through astrocytic hemichannels is sustained long after corticosterone treatment.”
3) Reviewer 2’s comment
When were Tukey´s and Dunnett´s tests used and what was the criteria to use one or the other?
Response:
Dunnett’s test for multiple comparisons was used to compare the means of different treatment groups against a reference group mean to determine if treatment significantly altered the outcome measure in question comparted the reference group. Tukey’s test was used to make comparisons between the means of every treatment group.
4) Reviewer 2’s comment
Carbenoxolone is a notoriously unspecific drug. In particular, it inhibits 11-beta-HSD2, an enzyme involved in converting glucocorticoids to inactive metabolites. This is particularly important in cells that co-express GR and MR Do astrocytes express HSD2? If so, could carbenoxolone alter the response to glucocorticoids?
Response:
The main point of using carbenoxolone in the current study was to non-selectively block connexins and pannexins to decrease release of HMGB1. To confirm a role of pannexin-1 on the release of HMGB1, 10Panx, a selective pannexin-1 inhibitor, was also used and demonstrated significant block of the release of HMGB1.
The Reviewer noted that carbenoxolone blocks 11-beta-HSD2. Based on a literature search, it is not known if 11-beta-HSD2 is expressed in astrocytes and whether 11-beta-HSD2 expression cycles over time, such as in ovary, is not known (Michael et al., 2003, Reproduction). Dexamethasone appears to increase 11-beta-HSD2 expression bronchial epithelial cells and is not significantly metabolized by 11-beta-HSD2 (Edwards et al., 1996, Steroids; Suzuki et al., 2003 Am J Respir Crit Care Med). Thus, a role of astrocytic 11-beta-HSD2, if it exists at all, on HMGB1 expression would be complex and require a separate study.
5) Reviewer 2’s comment
How do you normalize the western blots obtained from supernatant (Fig.1, Fig.2)? Data from whole-cell lysates could be used cells (Fig.4), but it is unclear if this was the way the experiments were performed.
Response:
To make comparison of the supernatant (HMGB1 released into the media) between treatment groups possible, equal volumes of conditioned media from equal number of cells was precipitated with TCA for western blotting.
Revised Material and methods:
Lines 104-106. “The conditioned media sample was spun at 1,000 g for 5 min to remove dead cells, and then equal volume of conditioned media from equal number of cells from each treatment group was precipitated with 15% Trichloroacetic Acid (TCA) on ice overnight.”
6) Reviewer 2’s comment
Pictures in Fig.3A do not seem to be representative of the data described in B. It would be nice to see a field with cells where HMGB1 is still nuclear and others with cytosolic localization.
Response:
Because immunocytochemistry of primary cultured astrocytes was performed at low cellular density to avoid overlapping astrocytes cell body, it was not possible to show a lot of cells in the same field at high magnification. Nonetheless, roughly equal numbers of DAPI positive nuclei were counted in vehicle and corticosterone-treated cultures. The comments have been added to Material and methods section:
Revised Material and methods
Lines 167-168. “The average of number of counting DAPI positive nuclei is about 70 in individual culture within an experiment.”
7) Reviewer 2’s comment
Regarding Fig.5 and Fig.6, why not follow the criteria from Fig.1, Fig.2 and Fig.4 and normalize data to basal expression of HMGB1?
Response:
Primary cultured astrocytes showed a slightly different basal expression of HMGB1 release depending on culture conditions. Thus, data within each condition were normalized to “vehicle treatment (corticosterone alone)” in Fig. 5 and Fig. 6.
8) Reviewer 2’s comment
Line 87: correct “carbenoxolne”, it should be “carbenoxolone”
9) Reviewer 2’s comment
Line 188: “confirm” should be “confirmed”
Response:
Spelling errors were corrected line 88 and line 186.